# Put Some Guts into It: Intestinal Organoid Models to Study Viral Infection

**DOI:** 10.3390/v12111288

**Published:** 2020-11-11

**Authors:** Inés García-Rodríguez, Adithya Sridhar, Dasja Pajkrt, Katja C. Wolthers

**Affiliations:** 1OrganoVIR Lab, Department of Medical Microbiology, Amsterdam UMC, Academic Medical Center, University of Amsterdam, 1100 AZ Amsterdam, The Netherlands; i.garciarodriguez@amsterdamumc.nl (I.G.-R.); a.sridhar@amsterdamumc.nl (A.S.); 2Department of Pediatrics Infectious Diseases, Emma Children’s Hospital, Amsterdam UMC, Academic Medical Center, University of Amsterdam, 1100 AZ Amsterdam, The Netherlands; d.pajkrt@amsterdamumc.nl

**Keywords:** intestinal organoid, enteroid, intestinal monolayer, Transwell^®^, Gut-on-a-Chip, Intestine-on-a-Chip, host-virus interactions, enteric virus

## Abstract

The knowledge about enteric viral infection has vastly increased over the last eight years due to the development of intestinal organoids and enteroids that suppose a step forward from conventional studies using cell lines. Intestinal organoids and enteroids are three-dimensional (3D) models that closely mimic intestinal cellular heterogeneity and organization. The barrier function within these models has been adapted to facilitate viral studies. In this review, several adaptations (such as organoid-derived two-dimensional (2D) monolayers) and original intestinal 3D models are discussed. The specific advantages and applications, as well as improvements of each model are analyzed and an insight into the possible path for the field is given.

## 1. Introduction

The intestinal epithelium forms a physical barrier between the body and the external environment. This polarized barrier is composed of a single cell layer with different cell types that execute several functions including nutrient uptake and protection from pathogens, such as enteric viruses [1,2]. These pathogens can alter the normal function of the epithelium in several ways such as cell death, disruption of tight junctions (TJ) between the cells, deregulation of the ion transport across the epithelium, or induction of inflammation by cytokine and chemokine responses [3].

Historically, host-virus interactions have been studied using cancer-derived or immortalized cell lines. In the context of human enteric viruses, the most used cell lines are Caco-2 and HT29 cells, both derived from colorectal adenocarcinoma. Apart from its conventional use as two-dimensional (2D) monolayers in standard culture plates, these cell lines can also be grown in permeable supports, such as Transwell^®^ inserts, where they can form a polarized monolayer with TJs and adherens junctions [4]. Such Transwell^®^ systems have been used to study several aspects of viral infection like polarity of infection [5,6], direct transcytosis [7] and via other cell types like lymphocytes [8,9], or viral entry [10,11]. However, despite many attempts, some viruses, such as human norovirus, cannot be cultured using standard 2D monolayer cultures, requiring alternative models for in-depth studies. One of these models is the rotating-wall vessel (RWV) bioreactors that allow the growth of three-dimensional (3D) cell spheres that can be subjected to physiological conditions of fluid-shear. With the use of this technology and an embryonic intestinal epithelial cell line INT-407, it was possible to culture human norovirus for the first time [12]. However, for this particular purpose, this model has not been proven to be robust across laboratories [13,14,15].

A turning point for in vitro enteric pathogen studies came in 2011 when a protocol for the culture of intestinal organoids derived from human pluripotent stem cells (PSCs) [16] was developed (see Box 1 for definition). In 2012, embryonic PSC-derived organoids were used to study infection of rotavirus clinical isolates [17]. After this first demonstration, more viruses were studied using this technology and specific models for different viral studies were developed (see [18] for details on specific viruses and [19] for an in-depth review of the models). An overview of the milestones that led to this development is provided in Figure 1.

Box 1Standard nomenclature as proposed by the members of the Intestinal Stem Cell Consortium (ISCC) [20].
**Organoid:** 3D structure formed by polarized epithelial cells and their underlying mesenchymal elements. Although they can be isolated from primary material, the most common way of generating organoids is by using embryonal or induced PSCs.**Enterosphere:** spherical structure composed of intestinal epithelial cells that appears as a rounded cyst.**Enteroid:** multilobulated structure with a lumen derived from an enterosphere, solely composed of epithelial elements.**Colonosphere:** spherical structure composed of colonic epithelial cells that appears as a rounded cyst.**Colonoid:** multilobulated structure derived from a colonosphere with a lumen derived from colonic epithelial cells.Even though several terms are defined by the ISCC, they have not been adopted globally [21] and some terms, such as colonoid, are not completely adopted in the virology field.


In this review, we aim to analyze the models that are currently being used to study human viral infections, assess which model is best for each application and discuss further developments that could enhance the utility of current models.

## 2. Organoids Are Superior In Vitro Models for Studying Host-Pathogen Interactions

The protocols for generating the organoids/enteroids have remained practically unaltered since their initial publications. Briefly, enteroids are generated from the intestinal crypts that contain stem cells. They can be isolated from the tissue using detergents and maintained with several factors: Wnt pathway agonists and epidermal growth factor (EGF) for intestinal proliferation, and Noggin to increase the crypt number. These crypts are subsequently grown in Matrigel^®^, a protein mixture that mimics the extracellular matrix (ECM) and allows for the maintenance of the 3D structure [22,23]. For their part, intestinal organoids are usually derived from PSCs and differentiated into the specific cell types of the intestine using a series of growth factors [16].

The main characteristics that make intestinal organoids/enteroids (Box 1) an attractive system to study viral-host interaction are depicted in Table 1. Intestinal organoids or enteroids mimic more closely the in vivo situation than cell lines. They usually contain multiple cell types that can be found in the intestine: stem cells, Paneth cells, enterocytes, goblet cells and enteroendocrine cells [22]; while cell lines consist of a unique cell type. Moreover, cell lines are usually derived from (human- or animal-derived) tumors or animal cells, or transformed with specific mutations that allow indefinite proliferation thus not representing the situation of a healthy tissue [24].

Several studies that used both cell lines and enteroids have specifically demonstrated the improvements that this new technology provides. Because of their healthy tissue origin, enteroids present an intact innate immune response. For example, interferon (IFN) pretreatment of enteroids inhibited adenovirus replication while the same treatment did not have an effect in cell lines [25]. Additionally, enteroid infection with echovirus 11 induced antiviral pathways that were not induced after Caco-2 cell infection [26]. Similarly, human astrovirus induced several IFN stimulated genes (ISGs) in enteroids of different regions of the intestine but not in the Caco-2 cell line [27].

Moreover, the use of intestinal organoids and enteroids has led to striking findings. For instance, it was discovered that many enteroviruses, including poliovirus type 1, express a new protein in the upstream open reading frame (termed UP). Viral growth studies in enteroids using several enterovirus strains that were knocked out for UP showed viral growth attenuation, while the same experiment on cell lines had no effect from UP knockout. This observation, together with an increase in replication after detergent treatment of the epithelium, suggests that the protein may have a role in membrane disruption enhancing viral release [28].

## 3. Organoids and Organoid-Derived Models

Several modifications to the organoids/enteroids described in Section 2 have been made posteriorly to improve their utility for viral studies. Using these modifications different models (Figure 2) have been designed which will be discussed below.

### 3.1. 3D Structure

Some models maintain the original 3D structure as this resembles the human organ best. However, due to its closed shape, access to the luminal side is only possible through microinjection [29], by cutting the intestinal organoids/enteroids in half [17,30] or by breaking them completely down followed by Matrigel^®^ re-embedding [25,31,32] or seeding on pre-coated plates [31,33,34,35,36,37]. The first two techniques are time-consuming and require specific laboratory equipment and skilled professionals. These problems can be overcome with the breaking-down technique. However, using the aforementioned technique, the intestinal organoids/enteroids suffer from mechanical stress during the breaking-down process and lose the polarity of the epithelium.

Even though the original 3D model is useful for specific studies, such as swelling assays that mimic the fluid secretion frequently occurring in the body during a diarrheal episode [38], the difference in size and shape within different intestinal organoids/enteroids (see Figure 2) hinders its standardization. In the context of infection studies, as it is not possible to analyze infection polarity, the closed 3D models are not suitable to study pathogenic cell entry and receptor binding. For these reasons, modified 2D monolayers derived from closed 3D models have proven very suitable.

### 3.2. 2D Monolayers

To overcome some of the disadvantages of the original 3D model, 2D epithelial monolayers derived from closed 3D intestinal organoid/enteroid models were designed. Briefly, the 3D intestinal organoids/enteroids are dissociated into single cells using enzymatic methods and the cells are seeded onto coated plates [39]. Coating of the plates helps cell attachment and can be performed with different proteins that mimic the ECM; Matrigel^®^ [39,40,41,42] and collagen [28,43,44,45] are the most frequently reported. This method allows for the generation of a polarized monolayer with an exposed apical side that facilitates infection. Additionally, because the culture is performed in conventional culture plates, it can be scaled up and used in high-throughput screening (HTS) studies [46]. As an example, the use of this system proved to be successful in evaluating neutralization assays with viruses (such as human norovirus) that do not infect cell lines [44].

Although the 2D monolayers derived from 3D intestinal organoids/enteroids have a major advantage for virology studies, there are some limitations. 2D monolayers are simplistic models that can lose some of the interactions present in the 3D structures as the cells within the closed 3D model may be organized in a different way. Moreover, while these 2D monolayers open a possibility to interact with the apical side of the epithelium they do not allow access to the basolateral side. Therefore, polarity of infection or translocation cannot be studied with this system.

### 3.3. 2D Monolayers on Transwell^®^ Inserts

The 2D monolayer on Transwell^®^ inserts uses an analogous approach as the 2D monolayer system, but instead of plating the single cells onto conventional cultureware, they are plated onto Transwell^®^ inserts. These inserts have a porous membrane that, when placed on a multi-well plate, generate two separated compartments. For this specific application, the intestinal cells (from the closed 3D intestinal organoids/enteroids) or crypts are seeded on the upper chamber of the Transwell^®^ creating a polarized monolayer with access to both the apical and the basolateral compartments (shown in Figure 2). Similar to the monolayers described in Section 3.2., the Transwell^®^ is coated with ECM-like proteins. In this model, collagen is the most used protein to mimic the ECM [27,47,48,49].

The main advantage of using this 2D monolayer on Transwell^®^ inserts system is the possibility to infect and sample on the apical and basolateral sides. Using this model together with a protocol that induces the generation of microfold (M) cells, intestinal cells that participate in mucosal immunity, it was possible to study transcytosis of reovirus. This virus was added to the apical compartment and viral particles could be detected on the basolateral compartment as early as 4 h post-infection while this phenomenon did not occur in the monolayers without induced M cells [49]. The differences in infection sites (polarity) within this system are shown for various enteroviruses using this system. While enterovirus A71 preferentially infects cells from the apical surface of the 2D monolayer on Transwell^®^ inserts system, echovirus 11 does so from the basolateral side [50].

Although this model can also be applied for HTS studies, its use is limited by cost and availability of formats (96-well formats are the smallest available size currently used). Furthermore, cell imaging on Transwell^®^ inserts can be complicated as antibodies can bind nonspecifically to the membrane resulting in high background signals. In some cases, authors opt to do imaging using chamber slides to overcome this issue [49,51].

### 3.4. Intestine-On-A-Chip

While the 3D closed intestinal organoid/enteroids and 2D monolayer systems exhibit major advantages for their use in virology, a common disadvantage is the lack of dynamics. Current 2D or 3D intestinal organoid/enteroids systems are static and lack the peristaltic movements characteristic of the intestine. Efforts with microfluidics devices to generate an Intestine-on-a-Chip model have been developed for the purpose of introducing shear fluid. These devices generate a microenvironment similar to the in vivo situation and allow for real-time monitoring [52]. An Intestine-on-a-Chip model using enteroids grown as monolayers showed a more in vivo representative transcriptomic profile as compared to 3D enteroids. Moreover, the intestinal monolayer in the chip was able to self-organize into villus-like protrusions and the specific cell characteristics were confirmed for the different cell types [53].

Although primary material-derived chips have never been used for viral analysis, a previous Gut-on-a-Chip model with Caco-2 cells was used to study Coxsackie B1 virus. After apical infection, there was a gradient production of cytopathic effect (CPE) due to the fluidic flow and progeny viral particles could be detected, proving the suitability of this model for viral studies [54].

Despite the advantages provided by this model it requires specific material and fluidics systems that are not available or accessible for most laboratories. Furthermore, whether the application of peristaltic flow has any inherent benefit for virology studies remains to be seen. Moreover, the possibilities for scale-up of HTS studies are limited and the life span of the cells in the chips is short [52].

## 4. Conclusions and Future Perspectives

The use of intestinal organoids/enteroids is not only suitable to study known enteric viral pathogens, but they also give us an opportunity to analyze mechanisms of disease in depth and to study host cell receptor binding, intracellular signaling, and antiviral compound testing. Moreover, intestinal organoids/enteroids provide a tool to rapidly unravel insights into the pathology of novel viruses such as the newly discovered severe acute respiratory syndrome coronavirus 2 (SARS-CoV-2). Several groups have used enteroid technology to understand its pathogenicity after detection of viral RNA in feces of infected patients [55,56,57,58]. Although all of them showed viral infection of human enterocytes, the translation back to the patient situation is still unknown. One study found that the released virus is inactivated with human colonic fluid [55], while another was able to isolate infectious particles from a positive fecal sample [58].

One of the main issues in adopting enteroids or enteroid-derived models is the availability of primary material. As healthy human tissue is obtained from surgeries or biopsies, access to healthy human tissue becomes increasingly difficult with a shift towards minimally invasive surgeries. One option to overcome these difficulties is to work with induced PSC-derived organoids, as these cells are commercially available. However, it usually takes over a month to generate mature organoids, while enteroids derived from human tissue can be generated within a few days [31]. Furthermore, PSC-derived organoids also contain mesenchymal elements and have a more fetal phenotype that could hamper the utility of the model. For instance, fewer cells were infected with rotavirus using intestinal organoids versus enteroids, probably due to a fetal phenotype [17,38]. Although both intestinal organoids and enteroids can be cryopreserved [59,60], long-term use of intestinal organoids is limited, while enteroids can be expanded indefinitely [24].

Another barrier is the high costs of culture due to expensive media and growth factors compared to cell lines. Further, organoid culture can be technically challenging and requires investment in skilled personnel to implement the technology.

Although all models discussed suppose a step forward from cell lines, the use of one or the other depends largely on its application. Several improvements are still needed to increase the possibilities of this technology. For example, the innate immune response is better recapitulated on these intestinal organoid/enteroid models in comparison to cell lines. However, immune cells are not present in the currently available models, limiting the study of more complex cellular responses such as the adaptive immune response. To overcome this limitation, complex co-culture systems are being developed, with placement of immune cells on the basolateral side of intestinal monolayers. Such a complex model showed the ability of macrophages to sense *Escherichia coli* on the apical side of the membrane, stretch out and phagocytose the bacteria [61]. These macrophages were isolated from human blood from a different donor than the epithelial monolayer. It is also possible to isolate intestinal immune cells after detachment of the epithelial mucosal layer. This way it will be possible to obtain both epithelial and immune cells from the same donor and co-culture them [62].

Another addition that could enhance the recapitulation of the intestine is the inclusion of the intestinal microbiota. It has been established that the host’s microbiota, its components or its metabolites can influence viral infection [63]. For instance, incubation of poliovirus with bacteria or bacterial surface polysaccharides increased infectivity in vitro by facilitating adherence to HeLa cells, and in vivo, as shown by a decreased number of deaths when germ-free animals were used [64]. One of the first studies including bacteria with human intestinal progenitor cells showed that the addition of *Lactobacillus rhamnosus*, a probiotic bacterium, increased cell proliferation and differentiation of Paneth cells [65]. Similarly, colonization of human intestinal organoids with non-pathogenic *E. coli* also increased proliferation, enterocyte maturation and mucus production [66]. Both studies suggest that the inclusion of bacteria may enhance differentiation and thus resemblance to the in vivo tissue. However, several technical difficulties like sustained co-culture and culture of anaerobic bacteria in an anaerobic environment need to be addressed [67].

In conclusion, the development of human enteroids and intestinal organoids has revolutionized our understanding of the human intestine, and more specifically of its interactions with enteric pathogens. Due to their close similarity to the in vivo situation, these models provide tools to discover new insights in disease mechanisms and thus to develop targeted treatments. Their superiority over conventional cell culture approaches has demonstrated the necessity for the virology field to switch to these more reliable models to study human disease.

## Figures and Tables

**Figure 1 viruses-12-01288-f001:**
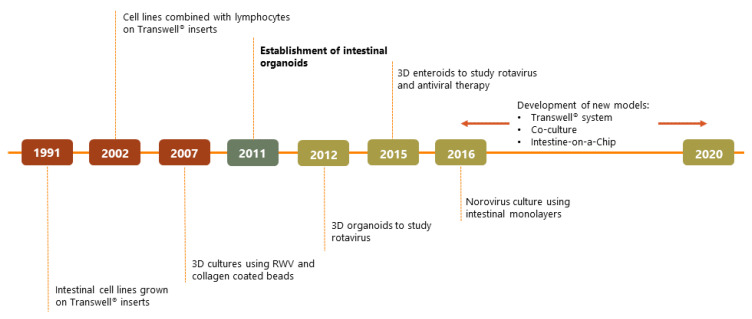
Timeline scheme depicting key points for enteric viral studies in vitro.

**Figure 2 viruses-12-01288-f002:**
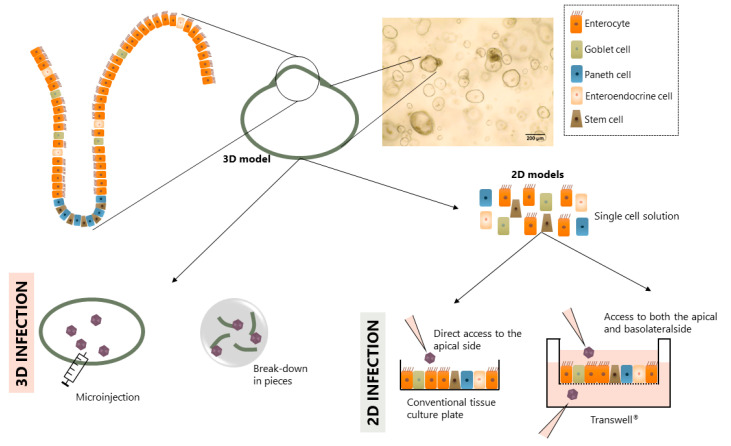
Schematic representation of the different models derived from intestinal organoids/enteroids applied for infection.

**Table 1 viruses-12-01288-t001:** Summary of organoid properties that enhance virology studies.

Organoid Property	Advantage to Virology
Cell heterogeneity and species specificity	Studies on factors crucial for viral pathogenesis in a representative species-specific physiological model, and on cell and tissue tropism.
Donor-specific characteristics	Organoids can be derived from donors of different ages and can be used to study preferential viral infection of certain age groups (child versus adult).
Scalability and high throughput	Opportunity to scale up for high throughput screening of antiviral strategies as organoids derive from stem cells which can proliferate indefinitely.

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
