# Peer review of "Put Some Guts into It: Intestinal Organoid Models to Study Viral Infection"

_viruses, 2020, doi:10.3390/v12111288_

Round 1
Reviewer 1 Report
The review titled “Put some guts into it: intestinal organoid models to study viral infection” submitted by García-Rodríguez et al discusses the use of stem-cell derived models of the gastrointestinal tract to study viral infections. This review introduces two models: iPSC/embryonic stem cell and tissue stem cell derived cultures and discusses how aspects such as plating format have influenced the advancement of viral pathogenesis. The use of organoids and enteroids to study common intestinal infections caused by rotavirus, norovirus, adenovirus and enterovirus is also discussed. Although the review is clearly written, it is not very well organized and does not reflect the current state of organoid research including using the most up to date terminology. In addition, important highlights are not discussed in an organized fashion. It is a very superficial overview/discussion and as such has limited value.
Revision suggestions:
Overall, this could benefit from distinguishing between differences in iPSC (defined by the authors ad organoids) and tissue stem cell (defined by the authors as enteroids) derived human cultures. The differences in these cultures should be established prior to the introduction of these cultures for use in studying enteric viral infections. The discussion of these cultures is not complete enough so that the reader really grasps what these cultures are and how these differences are relevant to the study of enteric infections. Once this is established, the review should highlight key enteric viruses in which major progress in the understanding of viral pathogenesis has been made using the organoid/enteroid cultures. This discussion should include a comprehensive overview of norovirus, rotavirus, enterovirus, astrovirus, adenovirus, and echovirus with each having its own paragraph. This will greatly improve the clarity and comprehensive nature of the review in addition to giving the reader a solid idea of the progress that has been made for each individual enteric virus over the broad overarching statements that lack sufficient support. Some mention needs to be made that there are differences in the iPSC and tissue stem cell derived organoid cultures and one needs to be aware of them as they could affect the interpretation of results with viral infections.
- Line 40-43 doesn’t make sense. If the model was irreproducible, then how could it be the first to demonstrate that human noroviruses were cultured for the first time? It either is or isn’t and the words contradict themselves here. This is confusing for someone not in the field.
- Line 45-46 and Figure 1. The mouse organoids were not a turning point for enteric pathogen studies. Rather it was the development of human intestinal epithelium cultures that significantly advanced the studies of enteric viral pathogens.
- Box 1 does not have complete or correct definitions of the different models.
- Table 1 is confusing. The addition of lines or shading for the rows would help clarify the relationships in the table.
- Section 3 should be moved earlier in the review as it is an introduction to the cultures and defines their origin and how they are made. The current placement makes the review seem unorganized and disjointed.
- There is no discussion of the role that the microbiome might play in the model or how it might reveal new information about viral pathogenesis. This seems an important concept to discuss.
- Lines 186-190 are important and should be discussed early. The placement contributes to the fragmented and disjointed nature of how the review flows and reads.
Author Response
We would like to thank the reviewer for taking the time to review our manuscript and highly appreciate the detailed feedback provided and suggestions. We have addressed their specific points below:
- Review comment: The review titled “Put some guts into it: intestinal organoid models to study viral infection” submitted by García-Rodríguez et al discusses the use of stem-cell derived models of the gastrointestinal tract to study viral infections. This review introduces two models: iPSC/embryonic stem cell and tissue stem cell derived cultures and discusses how aspects such as plating format have influenced the advancement of viral pathogenesis. The use of organoids and enteroids to study common intestinal infections caused by rotavirus, norovirus, adenovirus and enterovirus is also discussed. Although the review is clearly written, it is not very well organized and does not reflect the current state of organoid research including using the most up to date terminology. In addition, important highlights are not discussed in an organized fashion. It is a very superficial overview/discussion and as such has limited value.
- Our response: We have made some rearrangements in the text to facilitate the reading and further discussion points have been included. They will be detailed later on.
- Reviewer comment: Overall, this could benefit from distinguishing between differences in iPSC (defined by the authors ad organoids) and tissue stem cell (defined by the authors as enteroids) derived human cultures. The differences in these cultures should be established prior to the introduction of these cultures for use in studying enteric viral infections. The discussion of these cultures is not complete enough so that the reader really grasps what these cultures are and how these differences are relevant to the study of enteric infections.
- Our response: We thank the reviewer for this feedback. We have tried to further clarify this definition in Box 1, this box has also been cited in the introduction when first mentioned.
- Reviewer comment: Once this is established, the review should highlight key enteric viruses in which major progress in the understanding of viral pathogenesis has been made using the organoid/enteroid cultures. This discussion should include a comprehensive overview of norovirus, rotavirus, enterovirus, astrovirus, adenovirus, and echovirus with each having its own paragraph. This will greatly improve the clarity and comprehensive nature of the review in addition to giving the reader a solid idea of the progress that has been made for each individual enteric virus over the broad overarching statements that lack sufficient support.
- Our response: While we understand the reviewers point, we feel that the addition of overview on the different viruses is beyond the scope of this review. Our aim is to provide an introduction into organoids and facilitate the choice of the “right” enteric organoid model. Reviews on biological insights in enteric viruses through the use of organoids are already covered by several excellent reviews that are referred to in our text.
- Reviewer comment: Some mention needs to be made that there are differences in the iPSC and tissue stem cell derived organoid cultures and one needs to be aware of them as they could affect the interpretation of results with viral infections.
- Our response: Further discussion on this topic has been included in the discussion (second paragraph, lines 204-209).
- Reviewer comment: Line 40-43 doesn’t make sense. If the model was irreproducible, then how could it be the first to demonstrate that human noroviruses were cultured for the first time? It either is or isn’t and the words contradict themselves here. This is confusing for someone not in the field.
- Our response: We thank the reviewer for the feedback. We aimed to clarify the message in this sentence. We aimed to state that various laboratories tried to reproduce the culturing of the norovirus in this RWV system, but were not able to detect any norovirus replication.
- Reviewer comment: Line 45-46 and Figure 1. The mouse organoids were not a turning point for enteric pathogen studies. Rather it was the development of human intestinal epithelium cultures that significantly advanced the studies of enteric viral pathogens.
- Our response: Both the text and Figure 1 has been adapted accordingly.
- Reviewer comment: Box 1 does not have complete or correct definitions of the different models.
- Our response: We have included further definitions and clarifications in Box 1.
- Reviewer comment: Table 1 is confusing. The addition of lines or shading for the rows would help clarify the relationships in the table.
- Our response: Table 1 has been adjusted accordingly.
- Reviewer comment: Section 3 should be moved earlier in the review as it is an introduction to the cultures and defines their origin and how they are made. The current placement makes the review seem unorganized and disjointed.
- Our response: The first paragraph of Section 3 is now moved to Section 2 for a better structure of our paper as compared to the initial structure.
- Reviewer comment: There is no discussion of the role that the microbiome might play in the model or how it might reveal new information about viral pathogenesis. This seems an important concept to discuss.
- Our response: We thank the reviewer for this valuable suggestion. new paragraph discussing the role of the microbiome has been added in the discussion with new references.
- Reviewer comment: Lines 186-190 are important and should be discussed early. The placement contributes to the fragmented and disjointed nature of how the review flows and reads.
- Our response: As this is one of the key points of discussion when it comes to the implementation of this technology, we think it is better placed in the discussion.
Reviewer 2 Report
This is a well-written and helpful review of intestinal organoid and enteroid systems. It is largely a positive article and some of the technical challenges could be addressed as well for greater balance (for example expense, labor, low passage requirements when frozen). The authors could also discuss variation among cell populations that are beginning to be established by single cell RNA seq and genetic engineering of cells.
Author Response
We would like to thank the reviewer for taking the time to review our manuscript and highly appreciate the detailed feedback provided. We have addressed their specific points below:
- Reviewer comment: This is a well-written and helpful review of intestinal organoid and enteroid systems. It is largely a positive article and some of the technical challenges could be addressed as well for greater balance (for example expense, labor, low passage requirements when frozen).
- Our response: Further arguments on the technical challenges for the implementation of this technology has been included in the discussion (lines 207-212). We could not find specific references on the low passage requirements for freezing. In our hands, later passages (+25) can be frozen and restored. If the reviewer could point to a specific reference, we will be happy to include it.
- Reviewer comment: The authors could also discuss variation among cell populations that are beginning to be established by single cell RNA seq and genetic engineering of cells.
- Our response: As the target of our review is probably an audience outside the organoid field we thought getting into so much detail will dilute the message and focus of the review.